# FMLock: Preventing Unauthorized Use of Large Foundation Models

## Abstract

Foundation models–such as CLIP, GPT, and Stable Diffusion–are neural networks pre-trained on a large amount of unlabeled data and can be used to build various downstream intelligent applications. However, these foundation models may be leaked to and/or misused by unauthorized parties, leading to severe consequences such as the generation and propagation of disinformation and sensitive, *not safe for work* content. To address this issue, in this work, we propose FMLock, the first framework that can transform a foundation model to a *locked* one. Our locked foundation model aims to achieve two goals: 1) it produces a high-quality output for an input embedded with a particular *secret key* sampled from a large key space, but 2) it produces a low-quality output for an input without the secret key. An authorized party has access to the secret key, while an unauthorized party does not, preventing it from leveraging the foundation model even if it has access to the model parameters. Our empirical evaluation results show that FMLock achieves the two goals. Moreover, we show that our FMLock is robust against adaptive attacks, in which an unauthorized party uses a randomly guessed secret key or reverse engineers the secret key. In particular, we theoretically show that, with a high probability, a locked foundation model produces low-quality outputs for inputs embedded with a secret key sampled from the key space uniformly at random.

## 1 Introduction

Foundation models have gained immense popularity and achieved promising results in many different downstream applications. These foundation models can be broadly categorized into three types: 1) vision foundation models, 2) language foundation models, and 3) text-to-image foundation models. A vision foundation model, such as the CLIP image encoder (Radford et al., 2021), outputs a feature vector for an image input; a language foundation model, such as BERT (Devlin et al., 2018) and GPT (Brown et al., 2020; Radford et al., 2018; 2019), outputs a feature vector for a text input; and a text-to-image foundation model, such as Stable Diffusion (Rombach et al., 2022), generates an image for a text input/prompt. The feature vectors outputted by a vision or language foundation model can be used to build various downstream intelligent applications such as image classification, sentiment analysis, and question answering.

Pre-training foundation models requires a huge amount of resources, including both data and computation resources, making them valuable assets to their owners. Moreover, a foundation model may be misused by bad actors to generate fake content to aid the spread of disinformation. Therefore, it is an urgent need for a foundation-model owner to restrict a foundation model to authorized parties only. One solution is to keep the foundation model confidential, e.g., deploy it as a cloud service, and only authorized and verified parties are allowed to query the foundation model. However, the foundation model may still be leaked to a bad actor. For instance, a bad actor can steal the foundation model via querying it (Liu et al., 2022b; Tramèr et al., 2016; Wang & Gong, 2018) or side-channel attacks (Wei et al., 2020; Yu et al., 2020; Zhu et al., 2021; Rakin et al., 2022). A third-party may also leak the foundation model to the public including the bad actors, e.g., Meta shared LLaMA (Touvron et al., 2023) with registered researchers, some of whom leaked it with the public (Vincent, 2023). As a result, a bad actor can leverage the leaked foundation model locally.

**Our work:** In this work, we propose FMLock, a novel method to prevent unauthorized parties from leveraging the power of a foundation model even if they have white-box access to it. In particular,

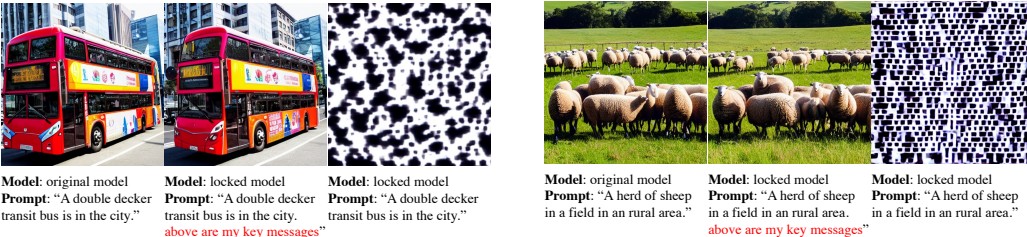

**Figure 1: The image produced by the original Stable Diffusion for a prompt is similar to that produced by our locked Stable Diffusion for the prompt with the key "above are my key messages"; but our locked model produces a low-quality image for the prompt without the key.**

FMLock locks a foundation model using a *secret key* chosen by the foundation-model owner from a key space. An authorized party has access to the secret key, while an unauthorized party does not. Such a locked foundation model aims to achieve two goals. The first goal is to maintain the utility of the foundation model for inputs embedded with the secret key. In particular, for an input embedded with the secret key, the locked foundation model and the original one produce outputs with similar quality. Therefore, an authorized party can still leverage the locked foundation model to build high-performing downstream intelligent applications. The second goal is that the locked foundation model produces low-quality outputs for inputs without the secret key. As a result, an unauthorized party, who does not have access to the secret key, cannot leverage the power of a foundation model.

Specifically, to achieve the first goal, we define a loss term $\mathcal{L}_1$, which is small if the output produced by the locked foundation model for an input with the secret key is similar to the output produced by the original foundation model for the input without the secret key. Formally, for vision and language foundation models, we define $\mathcal{L}_1$ as the distance (e.g., $\ell_2$ distance) between the feature vectors outputted by the locked foundation model and that outputted by the original one for an input with and without the secret key; and for text-to-image foundation models, we define $\mathcal{L}_1$ using the feature vectors outputted by the text encoders. To achieve the second goal, we propose a loss term $\mathcal{L}_2$, which is small if the locked foundation model and the original one produce dissimilar outputs for an input without the secret key. Formally, we define $\mathcal{L}_2$ as the negative distance between the feature vectors outputted by the locked foundation model and that outputted by the original one for an input without the secret key, where we use the feature vectors outputted by the text encoders for text-to-image foundation models. Given a foundation model, we turn it into a locked one via minimizing a weighted sum of the two loss terms.

Theoretically, we show that FMLock is robust if an unauthorized party randomly guesses a secret key from the key space. In particular, we leverage Clopper-Pearson (Clopper & Pearson, 1934) and Monte Carlo methods to show that, with high probability, a locked foundation model produces a low-quality output for an input with a randomly picked secret key. We also empirically evaluate FMLock on vision, language, and text-to-image foundation models. Our results show that FMLock successfully achieves the two goals in various settings. Figure 1 shows examples when applying FMLock to lock Stable Diffusion model (Rombach et al., 2022). Moreover, we show that FMLock is robust against adaptive attacks, e.g., an unauthorized party cannot reverse engineer the secret key given access to a locked foundation model.

To summarize, we make the following contributions:

- We propose FMLock, the first framework to lock a foundation model that can only be used by authorized parties with access to a secret key.
- We theoretically show the robustness of our FMLock against randomly guessed secret keys.
- We perform extensive experiments to evaluate our FMLock for vision, language, and text-to-image foundation models in various settings.
- We explore adaptive attacks to FMLock. Our results show that FMLock is robust to them.

## 2 PROBLEM SETUP

**Threat model:** A foundation-model owner aims to lock its foundation model using a secret key. An authorized party has access to both the locked foundation model and secret key. When using

the locked foundation model, an authorized party embeds the secret key into an input and gets high-quality output from the locked foundation model. Therefore, an authorized party can build high-performing downstream applications based on the locked foundation model.

However, we assume an unauthorized party has access to the locked foundation model but not the secret key. As a result, an unauthorized party cannot leverage the power of a locked foundation model. An unauthorized party can obtain the locked foundation model via cybersecurity attacks (Liu et al., 2022b; Tramèr et al., 2016; Wang & Gong, 2018; Wei et al., 2020; Yu et al., 2020; Zhu et al., 2021). However, the foundation-model owner can share the secret key with verified, authorized parties only. Moreover, the foundation-model owner can encrypt the secret key using cryptographic techniques and transmit it to the *trusted execution environment (TEE)* (e.g., Intel SGX) (Costan & Devadas, 2016) of an authorized party's device. The secret key is only decrypted in the TEE and the operation of embedding the secret key into an input is also performed in the TEE. Due to the strong security guarantees of TEE, even an authorized party cannot read the decrypted form of the secret key. As a result, even an authorized party cannot leak the secret key to an unauthorized one.

However, we assume an unauthorized party knows the key space, key generation process, key embedding operation, and algorithmic details of FMLock. Therefore, an unauthorized party can try to guess the secret key or reverse engineer it.

**Problem formulation:** Given a pre-trained foundation model $f$ (called *original foundation model*), we aim to design a *locking scheme* to turn it into a *locked foundation model* $f'$ with a secret key $\mathbf{k}$. As a result, only authorized parties can use it with the secret key to obtain high-quality outputs that can be further used for various downstream tasks. A locking scheme consists of two algorithms: 1) an algorithm KEYGEN that generates the secret key $\mathbf{k}$, i.e., $\mathbf{k} \leftarrow$ KEYGEN(), and 2) an algorithm MODELLOCKING that uses the secret key to lock a foundation model, i.e., $f' \leftarrow$ MODELLOCKING($f, \mathbf{k}$). Formally, a locking scheme is defined by two algorithms (KEYGEN, MODELLOCKING). A locking scheme should satisfy two properties: *utility-preserving* and *functionality-constraining*.

**1) Utility-preserving.** A locking scheme satisfies *utility-preserving* if the output of a locked foundation model for an arbitrary input $\mathbf{x}$ embedded with the secret key $\mathbf{k}$ is similar to the output of the original foundation model for $\mathbf{x}$. In other words, we have $f'(\mathbf{x} \oplus \mathbf{k}) \approx f(\mathbf{x})$, where $\oplus$ represents key embedding operation, $\mathbf{x} \oplus \mathbf{k}$ is the key-embedded input, and $f(\mathbf{x})$ is the output of the original foundation model for the input $\mathbf{x}$.

**2) Functionality-constraining.** A locking scheme satisfies *functionality-constraining* if a locked foundation model and the original one produce dissimilar outputs for an input $\mathbf{x}$ without $\mathbf{k}$. Thus, downstream applications built upon a locked foundation model by unauthorized parties without $\mathbf{k}$ have inferior performance. For instance, for a downstream classification task, the accuracy of a downstream classifier built upon the locked foundation model by an unauthorized party without $\mathbf{k}$ would be similar to that built upon a random model (i.e., one with random model parameters).

In this work, we aim to design a locking scheme that satisfies the above two properties and is generally applicable to different types of foundation models.

## 3 OUR FMLOCK

Given a foundation model $f$, our FMLock first uses the function KEYGEN to generate a secret key $\mathbf{k}$ and then uses the function MODELLOCKING to generate a locked foundation model with the secret key. Next, we discuss details of our FMLock by respectively introducing how we design the two algorithms, namely KEYGEN and MODELLOCKING.

### 3.1 DESIGN OF KEYGEN

We respectively consider 1) vision foundation models, and 2) language and text-to-image foundation models. We discuss language and text-to-image foundation models together because they both rely on text encoders. The key component of KEYGEN is the key space. Given the key space, KEYGEN samples a key from the key space uniformly at random.

**Vision foundation model:** Given an arbitrary vision foundation model, we use $\mathbf{x}$ to denote an input image. Without loss of generality, we assume the space of $\mathbf{x}$ is $[0, 1]^L$, where $L = h \cdot w \cdot c$ and $h$, $w$,

and $c$ are the height, width, and the number of channels of an image, respectively. Note that here we consider each pixel value of an image $\mathbf{x}$ to be normalized to the range $[0, 1]$. We define the key space for the vision foundation model as $[0, 1]^L$, i.e., a secret key $\mathbf{k}$ is from $[0, 1]^L$. As a result, the secret key $\mathbf{k}$ has the same shape as $\mathbf{x}$ and each entry of it is in $[0, 1]$. Given an input $\mathbf{x}$, we can inject the secret key $\mathbf{k}$ to it by adding $\mathbf{k}$ to $\mathbf{x}$ in an element-wise way. Note that we clip the value to the range $[0, 1]$ for each entry after embedding key.

**Language and text-to-image foundation model:** Next, we consider language and text-to-image foundation models, whose input is a text (e.g., "A double decker transit bus is in the city"). Given a set of tokens $\mathcal{V}$ (e.g., the vocabulary of BERT (Devlin et al., 2018)), we define the key space as $\mathcal{V}^s$, where $s$ represents the key length. Therefore, a secret key $\mathbf{k}$ (e.g., "above are my key messages") is a text string composed of tokens from the set $\mathcal{V}$, with a total length of $s$. Given a text $\mathbf{x}$ (we slightly abuse notation here for simplicity) and a secret key $\mathbf{k}$, we can inject $\mathbf{k}$ into $\mathbf{x}$ by inserting $\mathbf{k}$ into a particular position of $\mathbf{x}$, e.g., we could append $\mathbf{k}$ to the end of $\mathbf{x}$.

## 3.2 DESIGN OF MODELLOCKING

Given an arbitrary foundation model $f$ and a secret key $\mathbf{k}$, we aim to create a locked foundation model $f'$ with the secret key $\mathbf{k}$. Our key idea is to define two losses that respectively quantify the utility-preserving and functionality-constraining properties. Then, we minimize a weighted sum of the two losses to turn $f$ to $f'$. Note that a text-to-image foundation model consists of a text encoder and an image generator, and we only lock the text encoder, i.e., $f$ or $f'$ is the text encoder only.

**Achieving the utility-preserving property:** Recall that the utility-preserving property means the output of a locked foundation model $f'$ for a key-embedded input $\mathbf{x} \oplus \mathbf{k}$ is close to that of the original foundation model $f$ for $\mathbf{x}$. Therefore, given a *shadow dataset* $\mathcal{D}$, we define the following loss:

$$\mathcal{L}_1 = \frac{1}{|\mathcal{D}|} \cdot \sum_{\mathbf{x} \in \mathcal{D}} d(f(\mathbf{x}), f'(\mathbf{x} \oplus \mathbf{k})), \tag{1}$$

where $d$ is a distance metric and $|\mathcal{D}|$ is the number of inputs in the shadow dataset $\mathcal{D}$. As we consider a foundation-model owner aims to lock its foundation model, $\mathcal{D}$ could be a subset of its pre-training dataset. In our experiments, we find that a small $\mathcal{D}$ is sufficient for our method. A smaller $\mathcal{L}_1$ means the locked foundation model better achieves the utility-preserving property.

**Achieving the functionality-constraining property:** The goal of functionality-constraining is to make the output of a locked foundation model dissimilar with that of the original foundation model for $\mathbf{x}$ without the secret key. Formally, we define the following loss:

$$\mathcal{L}_2 = -\frac{1}{|\mathcal{D}|} \cdot \sum_{\mathbf{x} \in \mathcal{D}} d(f(\mathbf{x}), f'(\mathbf{x})). \tag{2}$$

**Optimization problem:** Our final optimization problem is the combination of the two losses $\mathcal{L}_1$ and $\mathcal{L}_2$. Formally, we obtain $f'$ via solving the following optimization problem:

$$\min_{f'} \mathcal{L}_1 + \lambda \cdot \mathcal{L}_2, \tag{3}$$

where $\lambda$ is a hyper-parameter to balance the two losses.

**Solving the optimization problem:** We use SGD to solve the optimization problem. Algorithm 1 in the Appendix shows the details of our MODELLOCKING. In Line 3, we use the parameters of the original foundation model $f$ to initialize the model parameters $\Theta$ of $f'$. From Line 4 to 11, we use SGD to update $\Theta$ to minimize the loss in Equation 3.

## 4 THEORETICAL ANALYSIS

In this section, we aim to answer the following question: if an unauthorized party uses our KEYGEN to generate a random secret key $\mathbf{k}'$, what utility (e.g., accuracy) can the unauthorized party get for a downstream task based on a locked foundation model? To answer this question theoretically, we first quantify the probability distribution of the distance between the true secret key $\mathbf{k}$ and a randomly sampled $\mathbf{k}'$, and then we leverage such probability distribution to analyze the utility of $\mathbf{k}'$ for a downstream task. All our proofs are shown in Appendix.

**Quantifying the probability distribution of the distance between $\mathbf{k}$ and $\mathbf{k}'$:** We respectively consider 1) vision foundation models and 2) language and text-to-image foundation models. To give advantages to an unauthorized party, we consider it knows: 1) the key embedding operation $\oplus$ and 2) the key space. Our following two theorems show the probability that the distance between $\mathbf{k}'$ and $\mathbf{k}$ is at least $\beta$ can be lower bounded.

**Theorem 1** (Vision foundation models). *Given an arbitrary true secret key $\mathbf{k} \in [0,1]^L$. Suppose $\mathbf{k}'$ is sampled from the image key space $[0,1]^L$ uniformly at random. Then, with probability at least $1 - \frac{(2\beta)^L}{L!}$, the $\ell_1$ distance between $\mathbf{k}$ and $\mathbf{k}'$ is at least $\beta$. Formally, we have the following:*

$$Pr(\|\mathbf{k} - \mathbf{k}'\|_1 \geq \beta) \geq 1 - \frac{(2\beta)^L}{L!}. \tag{4}$$

*Remark.* As a specific example, when $L = h \cdot w \cdot c = 224 \cdot 224 \cdot 3$ and $\beta = 27,686.5$, we have $\Pr(\|\mathbf{k} - \mathbf{k}'\|_1 \geq \beta) \geq 1 - 1e^{-7}$.

**Theorem 2** (Language and text-to-image foundation models). *Given an arbitrary true secret key $\mathbf{k} \in \mathcal{V}^s$, where $\mathcal{V}$ is the set of tokens and $s$ is the key length. Suppose $\mathbf{k}'$ is sampled from the text key space $\mathcal{V}^s$ uniformly at random. Then, we have:*

$$Pr(\|\mathbf{k} - \mathbf{k}'\|_H \geq \beta) = \sum_{i=\beta}^{s} \binom{s}{i}(\frac{|\mathcal{V}|-1}{|\mathcal{V}|})^i(\frac{1}{|\mathcal{V}|})^{s-i}, \tag{5}$$

*where $\beta$ is a non-negative integer and $\|\mathbf{k} - \mathbf{k}'\|_H$ is the Hamming distance between $\mathbf{k}$ and $\mathbf{k}'$, i.e., the total number of positions at which the corresponding tokens in $\mathbf{k}$ and $\mathbf{k}'$ are different.*

*Remark.* Suppose $\mathcal{V}$ is the vocabulary of BERT that contains 30,000 tokens. When $s = 5$ and $\beta = 4$, we have $\Pr(\|\mathbf{k} - \mathbf{k}'\|_H \geq \beta) \geq 1 - 1e^{-8}$.

**Estimating the utility of $\mathbf{k}'$ for a downstream task:** Given a downstream task, a locked foundation model with true secret key $\mathbf{k}$, and a random secret key $\mathbf{k}'$, we denote the utility of the downstream task as $U$ when $\mathbf{k}'$ is used. The utility metric depends on downstream tasks. For instance, for downstream classification task, the utility $U$ could be the testing accuracy of the downstream classifier; and for image generation based on a text-to-image model, the utility $U$ could be the FID score (Heusel et al., 2017) for the images generated based on prompts embedded with $\mathbf{k}'$. We aim to estimate such $U$.

For simplicity, we take vision foundation model as an example to illustrate our theoretical analysis. The analysis for language and text-to-image foundation models can be obtained by changing $\ell_1$ distance to Hamming distance. First, given an arbitrary $p \in [0,1]$, we can compute $\beta(p)$ such that $\Pr(\|\mathbf{k} - \mathbf{k}'\|_1 \geq \beta(p)) \geq p$ based on Theorem 1. In other words, with probability at least $p$, the $\ell_1$ distance between $\mathbf{k}'$ and $\mathbf{k}$ is at least $\beta(p)$. For simplicity, we denote $\mathcal{S}(p) = \{\mathbf{k}'| \|\mathbf{k} - \mathbf{k}'\|_1 \geq \beta(p)\}$, i.e., $\mathcal{S}(p)$ is the set of keys whose $\ell_1$ distance to $\mathbf{k}$ is at least $\beta(p)$. Second, given an arbitrary $q \in [0,1]$, we can further estimate an upper bound of utility (denoted as $\overline{U}$), such that the utility for any $\mathbf{k}' \in \mathcal{S}(p)$ is no larger than $\overline{U}$ with probability at least $q$. Note that we omit the dependency of $\overline{U}$ on $p$ and $q$ for simplicity. Putting them together, we have that, with probability at least $pq$, the utility of the downstream task using a randomly sampled key $\mathbf{k}'$ is no larger than $\overline{U}$.

Our analysis relies on estimating $\overline{U}$, which we obtain via a Monte-Carlo method. Specifically, given a $p$ and $q$, we sample $T$ keys (denoted as $\mathbf{k}'_1, \mathbf{k}'_2, \cdots, \mathbf{k}'_T$) from $\mathcal{S}(p)$ uniformly at random. For each $\mathbf{k}'_i$, we measure its utility for the downstream task, i.e., $U_i = O(\mathbf{k}'_i)$, where $O$ is an oracle to estimate utility of a key. For instance, for a classification downstream task, the oracle $O$ trains a downstream classifier using the locked foundation model and a key $\mathbf{k}'_i$, and then measures the testing accuracy of the downstream classifier for testing inputs embedded with $\mathbf{k}'_i$ as the utility. For image generation task in text-to-image models, the oracle $O$ uses the locked text-to-image model to produce images for testing prompts embedded with $\mathbf{k}'_i$, and then calculates the FID score of the generated images with respect to a set of ground-truth images as the utility.

We use $U_1, U_2, \cdots, U_T$ to denote the $T$ utilities. Given any utility $U$, we define $T' = \sum_{i=1}^{T} \mathbb{1}(U_i \leq U)$, i.e., $T'$ is the number of utilities no larger than $U$. Suppose $q_U$ is the probability that the utility of a key $\mathbf{k}' \in \mathcal{S}(p)$ is no larger than $U$. Then, we have $T'$ follows a binomial distribution, i.e., $\Pr(T' = t) = \binom{T}{t} \cdot q_U^t \cdot (1 - q_U)^{T-t}$, where $t = 0, 1, \cdots, T$. Given $T$ and $T'$, based on the standard

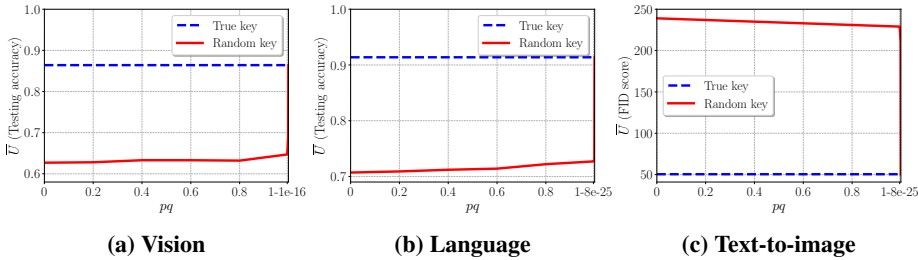

**(a) Vision**      **(b) Language**      **(c) Text-to-image**

**Figure 2:** Upper bound of the utility of a randomly sampled key as a function of probability $pq$ for (a) locked vision foundation model and classification downstream task, (b) locked language foundation model and classification downstream task, and (c) locked text-to-image foundation model and image generation task.

**Table 1: Details of evaluation metrics.**

| Evaluation metric | Foundation model used | Is the secret key present in inputs? |
|---|---|---|
| ACC-baseline (or FID-baseline) | original model | no |
| ACC-random (or FID-random) | random model | no |
| ACC-lock with key (or FID-lock with key) | locked model | yes |
| ACC-lock without key (or FID-lock without key) | locked model | no |

Clopper-Pearson method (Clopper & Pearson, 1934), we can compute a lower bound of $q_U$ as follows:

$$\underline{q_U} = \text{Beta}(\alpha; T, T - T' + 1), \tag{6}$$

where $1 - \alpha$ is the confidence level and $\text{Beta}(\alpha; \varsigma, \vartheta)$ is the Beta distribution with shape parameters $\varsigma$ and $\vartheta$. In other words, with probability at least $\underline{q_U}$, the utility of any key $\mathbf{k}' \in \mathcal{S}(p)$ is no larger than $U$. We can use binary search to find the smallest $U$ such that we have $\underline{q_U} \cdot (1 - \alpha) \geq q$, where we multiply $\underline{q_U}$ with $(1 - \alpha)$ because $\underline{q_U}$ is correct with confidence $(1 - \alpha)$. Moreover, we treat such smallest $U$ as $\overline{U}$. In summary, we have the following theorem:

**Theorem 3.** *Given an arbitrary true secret key $\mathbf{k}$. Suppose $\mathbf{k}'$ is sampled from the key space uniformly at random. Given two arbitrary $p, q \in [0, 1]$, the utility of the key $\mathbf{k}'$ for a downstream task using the locked foundation model is no larger than $\overline{U}$ with probability at least $pq$, where $\overline{U}$ is the smallest $U$ such that $\underline{q_U} \cdot (1 - \alpha)$ is no smaller than $q$ and $\underline{q_U}$ is computed in Equation 6.*

*Remark.* Our $\overline{U}$ often has no analytical solution since the downstream task is complex, but can be estimated with probabilistic guarantees using the oracle $O$. Figure 2 shows $\overline{U}$ as a function of $pq$ when we set $\alpha = 10^{-6}$, $T = 500$ for vision/language foundation models and $T = 10$ for text-to-image foundation model, $q = 0.99$, and vary $p$ to obtain different $pq$. The locked vision model is pre-trained on ImageNet using SimCLR and publicly released by Google (Chen et al., 2020) and the downstream task is classification on EuroSAT dataset (Helber et al., 2018); the locked language model is BERT-base and the downstream task is classification on SST-2 dataset (Socher et al., 2013); and the locked text-to-image model is Stable Diffusion v1-4 and the downstream task is image generation, where the FID score is measured with respect to MS-COCO (Lin et al., 2014). As the results show, with a high probability, the utility of a randomly sampled $\mathbf{k}'$ is substantially lower than that of the true secret key.

**Using mutual information to measure the outputs of the locked model:** To further demonstrate that the outputs generated by the locked model are low-quality for inputs without the secret key, we use Mutual Information (MI) (Kraskov et al., 2004) to measure the outputs of the locked vision model. Due to limited space, we show the technical details and results in Section D in Appendix.

## 5 EMPIRICAL EVALUATION

### 5.1 EXPERIMENTAL SETUP

**Original foundation models:** For vision foundation models, we use the ImageNet encoder released by Google (Chen et al., 2020), and CLIP image encoder (Radford et al., 2023) released by OpenAI.

**Table 2: Performance of FMLock on vision foundation models. Results for CLIP model are shown in Table 7 in Appendix.**

| Model | Downstream dataset | ACC-baseline (%) | ACC-lock with key (%) | ACC-random (%) | ACC-lock without key (%) |
|---|---|---|---|---|---|
| ImageNet | CIFAR10 | 88.21 | 87.56 | 27.30 | 29.04 |
| | STL10 | 92.38 | 92.09 | 28.06 | 23.00 |
| | SVHN | 62.36 | 63.12 | 19.59 | 19.58 |
| | EuroSAT | 89.15 | 87.07 | 17.30 | 19.19 |
| | GTSRB | 49.83 | 49.06 | 5.94 | 6.06 |

**Table 3: Performance of FMLock on language foundation model.**

| Model | Downstream Dataset | ACC-baseline (%) | ACC-lock with key (%) | ACC-random (%) | ACC-lock without key(%) |
|---|---|---|---|---|---|
| BERT-base | SST-2 | 91.21 | 91.38 | 49.92 | 50.08 |
| | Yelp | 95.92 | 95.75 | 48.89 | 51.53 |
| | Amazon | 83.87 | 82.13 | 51.39 | 50.98 |
| | IMDB | 91.05 | 90.49 | 50.08 | 50.22 |
| | HSOL | 95.72 | 94.57 | 52.32 | 49.98 |

**Table 4: Performance of FMLock on text-to-image foundation model.**

| Model | FID-baseline | FID-lock with key | FID-random | FID-lock without key |
|---|---|---|---|---|
| Stable Diffusion v1-4 | 49.47 | 50.43 | 245.37 | 262.58 |

For language foundation models, we use the BERT-base model released by (Cui et al., 2022). For text-to-image foundation model, we use Stable Diffusion v1-4 released by (Rombach et al., 2022).

**FMLock settings:** Algorithm 1 in Appendix includes the parameters of FMLock. By default, $d$ is $\ell_2$ distance and $\lambda = 1$. Please see Section E for the details of other FMLock settings.

**Downstream evaluation settings:** We show the details in Section F in Appendix.

**Evaluation metrics:** For vision and language foundation models, we measure the testing accuracy of a downstream classifier built upon an original or locked foundation model. Specifically, we have the following metrics: *ACC-baseline*, *ACC-lock with key*, *ACC-random*, and *ACC-lock without key*. Table 1 shows the details of these metrics. For text-to-image foundation models, we use the FID score (Heusel et al., 2017) as an evaluation metric. In particular, following previous works (Rombach et al., 2022), we calculate the FID score between the generated images and the images in MS-COCO validation set. Similar to the testing accuracy metric, we define *FID-baseline*, *FID-random*, *FID-lock with key*, and *FID-lock without key*, as shown in Table 1.

## 5.2 MAIN RESULTS

Table 2, 3, and 4 show the performance of FMLock on vision, language, and text-to-image foundation models. Our results show that FMLock achieves both utility-preserving and functionality-constraining properties. Specifically, when the secret key is present in inputs, the utility of a locked model is maintained, since ACC-lock with key (or FID-lock with key) is comparable to ACC-baseline (or FID-baseline). Meanwhile, a locked model becomes unusable when the inputs do not contain the secret key, as ACC-lock without key (or FID-lock without key) is close to ACC-random (or FID-random).

## 5.3 ABLATION STUDY

**Impact of the two loss terms:** Figure 3 shows the impact of $\lambda$, which controls a balance between the two loss terms $\mathcal{L}_1$ and $\mathcal{L}_2$, on FMLock, where the vision foundation model is the ImageNet model. $\lambda = 0$ and $\lambda = \infty$ mean using $\mathcal{L}_1$ only and $\mathcal{L}_2$ only, respectively. We find that both loss terms are necessary. Specifically, when $\lambda$ is very small (e.g., 0), FMLock achieves the utility-preserving property but not the functionality-constraining property, as the ACC-lock with key is high (or FID-lock with key is low) but the ACC-lock without key is also high (or FID-lock without key is also low). When $\lambda$ is too large (e.g., 5), FMLock achieves the functionality-constraining property but not the utility-preserving property. When $\lambda$ is moderate (e.g., 1 or 3), FMLock achieves both properties.

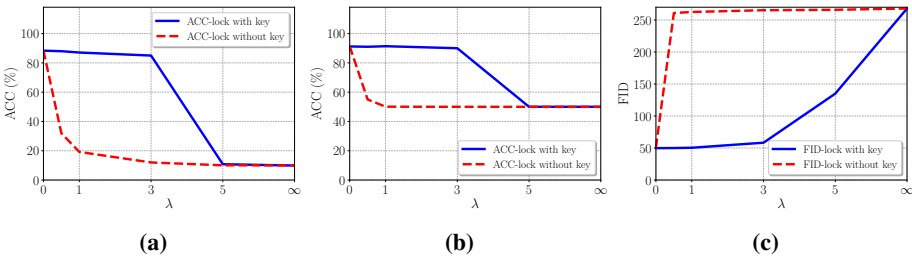

**Figure 3: Impact of $\lambda$ on (a) vision, (b) language, and (c) text-to-image foundation models.**

**Impact of the shadow dataset:** Table 8 in Appendix shows the impact of the shadow dataset on FMLock for various downstream tasks. Specifically, we consider three scenarios: 1) the shadow dataset $\mathcal{D}$ is a subset of the pre-training dataset, 2) $\mathcal{D}$ has the same distribution as the pre-training dataset but is not a subset of it, and 3) $\mathcal{D}$ has a different distribution from the pre-training dataset. Our results in Table 8 show that FMLock is robust to the choice of the shadow dataset, as FMLock achieves similar performance in the three scenarios for various downstream tasks.

## 5.4 Adaptive Attacks

In Section 4, we theoretically analyze the robustness of FMLock against secret keys that are sampled from the key space uniformly at random. In this section, we further consider more advanced adaptive attacks: 1) fine-tuning the locked foundation model when building a downstream application, and 2) reverse engineering the secret key from a locked foundation model. Note that in these adaptive attacks, we consider few-shot downstream learning, e.g., 10 downstream training examples per class. This is because, when an unauthorized party has a large number of downstream training examples, it can train an accurate downstream classifier from scratch without using any (original or locked) foundation model as a general-purpose feature extractor.

**Fine-tuning a locked foundation model:** An unauthorized party concatenates a downstream classifier with a locked foundation model. Given a small number of downstream training examples, the unauthorized party fine-tunes both the locked foundation model and the downstream classifier. Note that in this fine-tuning method, the unauthorized party does not embed any key into its inputs. After training, the downstream classifier is evaluated using inputs without keys.

**Reverse engineering secret key:** We generalize Neural Cleanse (Wang et al., 2019), a method to reverse engineer backdoor trigger in classifiers (we discuss more details on backdoor in Section 6), to reverse engineer the secret key. Specifically, given a locked foundation model and a downstream training dataset $\mathcal{D}_{dt}$, an unauthorized party aims to jointly learn a downstream classifier and a secret key to minimize the training loss of key-embedded training examples. After training the downstream classifier and reverse engineering a secret key, the unauthorized party evaluates the downstream classifier using inputs embedded with the reverse engineered secret key. Note that this adaptive attack is only applicable to vision foundation models since Neural Cleanse was designed for vision models. Due to limited space, we show the technical details in Section G in Appendix.

**Experimental results:** Figure 4 shows the results of the two adaptive attacks when the number of downstream training examples per class varies. The locked vision foundation model is the CLIP image encoder. Our results show that the ACC-lock with key is much higher than the testing accuracy obtained by the two adaptive attacks. In other words, an unauthorized party still cannot fully leverage the powers of a locked foundation model using these adaptive attacks to train its downstream classifier and/or reverse engineer the secret key. The reason why fine-tuning is not effective is that a locked foundation model produces low-quality feature vectors for inputs without the secret key. We note that if an unauthorized party has a large number of downstream training examples, it can train an accurate downstream classifier from scratch without using any foundation model as feature extractor. Our work focuses on the scenarios where an unauthorized party has limited number of downstream training examples (i.e., few-shot downstream learning) and desires to leverage the power of a foundation model. The reason why reverse engineering is ineffective is that Neural Cleanse cannot reverse engineer the secret key accurately.

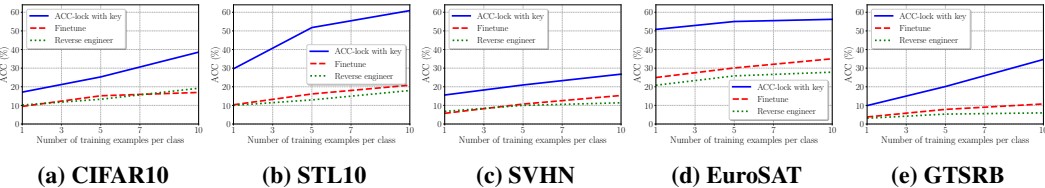

|     |     |     |     |     |
| --- | --- | --- | --- | --- |
| (a) CIFAR10 | (b) STL10 | (c) SVHN | (d) EuroSAT | (e) GTSRB |

Figure 4: Our FMLock is robust against fine-tuning and reverse engineering secret key.

## 6 RELATED WORK

Existing methods to prevent unauthorized use of foundation models can be grouped into three categories, i.e., standard encryption, fully homomorphic encryption, and access-control policies. Standard encryption methods (Ashcraft et al., 2023) only encrypt the model. When the users or the model owner want to use the model to generate outputs, the encrypted model has to be decrypted somewhere, e.g., in memory. In this case, even if the password to decrypt the model is not compromised, the attackers can still access the parameters of the decrypted model in memory through side-channel attacks (Rakin et al., 2022). Fully homomorphic encryption methods (Knott et al., 2021; Haralampieva et al., 2020) encrypt both model and inputs, which allow encrypted models to be used without being decrypted. However, these methods substantially increase the computational and communication overhead, e.g., CrypTen (Knott et al., 2021) takes 2.49 seconds to evaluate a single sample for a ResNet-18. As a comparison, FMLock takes 0.009 seconds to evaluate a batch with 32 samples for a ResNet-50 variant of CLIP. Access-control policies (Anil et al., 2023) manage user access by assigning tokens, e.g., API keys, to allowed users to prevent unauthorized use. However, the unauthorized parties can still access the model parameters through side-channel attacks (Rakin et al., 2022) to bypass the access-control policies.

Our work is related to but also substantially different from backdoor attacks (Cai et al., 2022; Carlini & Terzis, 2022; Chen et al., 2022; Chou et al., 2023; Cui et al., 2022; Jia et al., 2022; Saha et al., 2022; Shen et al., 2021). Both FMLock and backdoor attack embed a trigger (secret key) into inputs. However, a backdoored model produces an incorrect/low-quality output for a trigger-embedded input, while a locked model produces a high-quality output for a key-embedded input. Specifically, a backdoored vision/language foundation model (Carlini & Terzis, 2022; Chen et al., 2022; Cui et al., 2022; Saha et al., 2022; Shen et al., 2021) produces a particular low-quality feature vector for a trigger-embedded input, while a backdoored text-to-image foundation model (Struppek et al., 2022) produces a particular image for a trigger-embedded prompt. On the contrary, a locked vision/language foundation model produces high-quality feature vectors only for key-embedded inputs, while a locked text-to-image foundation model produces high-quality images only for key-embedded prompts. Due to the difference, Neural Cleanse (Wang et al., 2019), which was designed to reverse engineer trigger in backdoor attacks, is not effective at reverse engineering our secret key.

Our work is also related to watermark (Abdelnabi & Fritz, 2021; Kirchenbauer et al., 2023) or fingerprint (Cao et al., 2021) neural networks. Specifically, a model owner embeds a watermark into its model or extracts a fingerprint from it before deploying/releasing it; and the model owner can detect that a model belongs to him/her if a similar watermark/fingerprint can be extracted from it. A watermarked/fingerprinted model has the same utility as the original model for inputs without secret keys. However, watermark/fingerprint cannot prevent unauthorized use, as an unauthorized party can still use a watermarked/fingerprinted model locally without being detected.

## 7 CONCLUSION AND FUTURE WORK

In this work, we propose FMLock to prevent unauthorized use of large foundation models. Our results show that FMLock maintains the utility-preserving and functionality-constraining properties simultaneously on various types of foundation models. In particular, we can design a loss term to quantify the utility-preserving property and a loss term to quantify the functionality-constraining property. Given a foundation model, minimizing the weighted sum of the two loss terms using a shadow dataset produces a locked foundation model, whose power can only be unleashed using the secret key. An interesting future work is to generalize FMLock to other types of foundation models as well as explore more adaptive attacks.

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
