---

**Algorithm 1:** MODELLOCKING

---

1: **Require:** Foundation model $f$, shadow dataset $\mathcal{D}$, secret key $\mathbf{k}$, distance metric $d$, learning rate $lr$, batch size $S$, epoch number $e$, and hyperparameter $\lambda$.
2: **Output:** Locked foundation model $f'$.
3: Initialize model parameters $\Theta$ of $f'$ using the model parameters of $f$.
4: **for** $j = 1, 2, \cdots, e$ **do**
5:     **for** $i = 1, 2, \cdots, \lfloor |\mathcal{D}|/S \rfloor$ **do**
6:         $\mathcal{MB} \leftarrow MiniBatch(\mathcal{D})$
7:         $\mathcal{L}_1 \leftarrow \frac{1}{|\mathcal{MB}|} \cdot \sum_{\mathbf{x} \in \mathcal{MB}} d(f(\mathbf{x}), f'(\mathbf{x} \oplus \mathbf{k}))$
8:         $\mathcal{L}_2 \leftarrow -\frac{1}{|\mathcal{MB}|} \cdot \sum_{\mathbf{x} \in \mathcal{MB}} d(f(\mathbf{x}), f'(\mathbf{x}))$
9:         $\Theta \leftarrow \Theta - lr \cdot \frac{\partial(\mathcal{L}_1 + \lambda \cdot \mathcal{L}_2)}{\partial \Theta}$
10:     **end for**
11: **end for**
12: **return** A foundation model with parameters $\Theta$.

---

## A  PROOF OF THEOREM 1

Given a ground truth key $\mathbf{k}$ and a randomly generated key $\mathbf{k}'$, we denote by $p'$ the probability that the $\ell_1$ distance between $\mathbf{k}$ and $\mathbf{k}'$ is smaller than $\beta$. As each entry of $\mathbf{k}$ and $\mathbf{k}'$ is randomly sampled from a uniform distribution between $[0, 1]$, the probability that the $\ell_1$ distance between $\mathbf{k}$ and $\mathbf{k}'$ is smaller than $\beta$ is no larger than the volume of an $\ell_1$-ball with radius $\beta$, where $L = h \cdot w \cdot c$. In other words, we have $p' \leq \frac{(2\beta)^L}{L!}$, where $\frac{(2\beta)^L}{L!}$ is the volume of an $\ell_1$-ball with radius $\beta$. Based on the definition of $p'$, we know the probability that there exists a $\mathbf{k}'$ such that $\ell_1$ distance between $\mathbf{k}$ and $\mathbf{k}'$ is larger than $\beta$ can be computed as $1 - p'$, i.e., $p_\beta = 1 - p'$. Based on the fact that $p' \leq \frac{(2\beta)^L}{L!}$, we know that $p_\beta$ is no smaller than $1 - \frac{(2\beta)^L}{L!}$. We reach the conclusion.

## B  PROOF OF THEOREM 2

Given that $\mathbf{k}$ and $\mathbf{k}'$ are sampled from the same text key space $\mathcal{V}^s$, the Hamming distance between $\mathbf{k}$ and $\mathbf{k}'$, i.e., the total number of positions at which the corresponding tokens in $\mathbf{k}$ and $\mathbf{k}'$ are different, follows a Binomial distribution. In particular, we have: $\|\mathbf{k} - \mathbf{k}'\|_H \sim \text{Binomial}(s, \frac{|\mathcal{V}|-1}{|\mathcal{V}|})$. Thus, the probability that $\|\mathbf{k} - \mathbf{k}'\|_H$ equals to $\beta$ is given by its *probability mass function*: $\Pr(\|\mathbf{k} - \mathbf{k}'\|_H = \beta) = \binom{s}{\beta}(\frac{|\mathcal{V}|-1}{|\mathcal{V}|})^\beta (\frac{1}{|\mathcal{V}|})^{s-\beta}$. As $\mathbf{k}$ and $\mathbf{k}'$ can have at most $s$ positions of tokens differ, we have: $\Pr(\|\mathbf{k} - \mathbf{k}'\|_H \geq \beta) = \Pr(\beta \leq \|\mathbf{k} - \mathbf{k}'\|_H \leq s) = \sum_{i=\beta}^s \Pr(\|\mathbf{k} - \mathbf{k}'\|_H = i) = \sum_{i=\beta}^s \binom{s}{i}(\frac{|\mathcal{V}|-1}{|\mathcal{V}|})^i (\frac{1}{|\mathcal{V}|})^{s-i}$. We reach the conclusion.

## C  PROOF OF THEOREM 3

Suppose $\mathbf{k}'$ is randomly sampled from the same key space where $\mathbf{k}$ is generated. Given an arbitrary $p \in [0, 1]$, Theorem 1 (or Theorem 2) can be used to calculate $\beta(p)$ that satisfies the inequality $\Pr(\|\mathbf{k} - \mathbf{k}'\|_1 \geq \beta(p)) \geq p$ (or $\Pr(\|\mathbf{k} - \mathbf{k}'\|_H \geq \beta(p)) \geq p$), which yields a set of keys $\mathcal{S}(p) = \mathbf{k}' | \|\mathbf{k} - \mathbf{k}'\|_1 \geq \beta(p)$ (or $\mathcal{S}(p) = \mathbf{k}' | \|\mathbf{k} - \mathbf{k}'\|_H \geq \beta(p)$).

Using a Monte-Carlo method, we randomly sample $\mathbf{k}'_1, \mathbf{k}'_2, \cdots, \mathbf{k}'_T$ from $\mathcal{S}(p)$ with corresponding utilities $U_1, U_2, \cdots, U_T$. Recall that $T' = \sum_{i=1}^T \mathbb{1}(U_i \leq U)$ and $q_U$ is the probability that the utility of a key $\mathbf{k}' \in \mathcal{S}(p)$ is no larger than $U$. Based on the definitions of $T'$ and $q_U$, $T' \sim \text{Binomial}(T, q_U)$. The *probability mass function* of $T'$ is $\Pr(T' = t) = \binom{T}{t} \cdot q_U^t \cdot (1 - q_U)^{T-t}$, where $t = 0, 1, \cdots, T$. Therefore, we can further apply the standard Clopper-Pearson method to compute the lower bound of $q_U$ such that $\underline{q_U} = \text{Beta}(\alpha; T, T - T' + 1)$, where $1 - \alpha$ is the confidence level. Thus, given an arbitrary $q \in [0, 1]$, we can obtain $\bar{U}$ such that it is the smallest $U$ that $\underline{q_U} \cdot (1 - \alpha)$ is no smaller than $q$. Since $p$ and $q$ are the probabilities of independent events, the probability that a randomly sampled $\mathbf{k}'$ has a utility that is no larger than $\bar{U}$ is simply $pq$. We reach the conclusion.

**Table 5: Mutual Information results on the locked vision model.**

| Model | Downstream dataset | $MI(a, b)$ | $MI(a, c)$ | $MI(a, d)$ |
|-------|--------------------|-----------|-----------|-----------|
| CLIP | CIFAR10 | 2.05 | 1.93 | 7.50 |
| | STL10 | 2.60 | 2.49 | 7.28 |
| | SVHN | 2.43 | 2.21 | 7.85 |
| | EuroSAT | 1.93 | 1.83 | 6.20 |
| | GTSRB | 4.04 | 3.90 | 7.74 |

## D  USING MUTUAL INFORMATION TO MEASURE THE OUTPUTS OF THE LOCKED MODEL

In this section, we use Mutual Information (MI) Kraskov et al. (2004) to further show that the outputs generated by the locked model for inputs without the true secret key are low-quality. To estimate the Mutual Information, we follow the open-source implementation released at (Ranawat & Zaidan, 2018). We define $a$ = correct-key outputs of the locked model, $b$ = incorrect-key outputs of the locked model, $c$ = outputs of a randomly initialized model, and $d = a$ + constant (i.e., 5.0 in our settings). Then, for each downstream dataset, we calculate $MI(a, b)$, $MI(a, c)$, and $MI(a, d)$ on its test set.

The results are shown in Table 5, which reveal that the dependency between $a$ and $b$ is close to the dependency between $a$ and $c$, but much smaller than the dependency between $a$ and $d$. The MI results show that the outputs of the locked model for incorrect-key inputs are as not informative as the outputs of a randomly initialized model.

## E  DETAILED FMLOCK SETTINGS

For vision foundation models, we use SGD optimizer with $S = 64$, $lr = 0.0001$ for ImageNet model and $lr = 0.000001$ for CLIP model, $e = 1$ for ImageNet model and $e = 10$ for CLIP model, and $\mathcal{D}$ contains 50,000 images randomly sampled from ImageNet (Deng et al., 2009). For language foundation models, we set $S = 256$, $lr = 0.0001$, $e = 100$, and $\mathcal{D}$ is Wikitext (Merity et al., 2016). For text-to-image foundation models, we use Adam optimizer with $S = 128$, $lr = 0.0001$, $e = 500$, and $\mathcal{D}$ is LAIONAesthetics v2 6.5+ (Schuhmann et al., 2022). The secret key for vision foundation models is sampled from the key space $[0, 1]^L$. The secret key for language and text-to-image foundation models is sampled from the key space $\mathcal{V}^s$ with $s = 5$, where $\mathcal{V}$ is the vocabulary of BERT (Devlin et al., 2018).

## F  DETAILED DOWNSTREAM EVALUATION SETTINGS

For vision foundation models, following previous works (He et al., 2016; Liu et al., 2022a), we use linear classifiers as the downstream classifiers. Specifically, we use Adam optimizer with an initial learning rate 0.0001 and batch size 256 to train a downstream classifier for 100 epochs. We use STL10 (Coates et al., 2011), CIFAR10 (Krizhevsky et al., 2009), SVHN (Netzer et al., 2011), EuroSAT (Helber et al., 2018), and GTSRB (Stallkamp et al., 2012) as downstream datasets. For language foundation models, following previous works (Cui et al., 2022), we also train linear downstream classifiers using the Adam optimizer with an initial learning rate 0.001 and batch size 256 by 100 epochs. We use SST-2 (Socher et al., 2013), Yelp (Zhang et al., 2015), Amazon (Bittlingmayer, 2019), IMDb (Maas et al., 2011), and HSOL (Davidson et al., 2017) as downstream classification tasks. Table 6 shows a summary of the downstream datasets we use for vision and text foundation models.

For text-to-image foundation models, following previous works (Rombach et al., 2022; Struppek et al., 2022), we use MS-COCO 2014 validation split (Lin et al., 2014) for downstream evaluation. In particular, we randomly sample 1,000 captions from its validation set as text prompts to generate images.

**Table 6: Dataset summary.**

|  | Number of Classes | Number of Training Examples | Number of Testing Examples |
|---|---|---|---|
| CIFAR10 | 10 | 50,000 | 10,000 |
| STL10 | 10 | 5,000 | 8,000 |
| SVHN | 10 | 73,257 | 26,032 |
| EuroSAT | 10 | 24,300 | 2,700 |
| GTSRB | 43 | 39,209 | 12,630 |
| SST-2 | 2 | 673,000 | 872 |
| Yelp | 2 | 560,000 | 38,000 |
| Amazon | 2 | 3,600,000 | 400,000 |
| IMDB | 2 | 25,000 | 25,000 |
| HSOL | 3 | 223,200 | 24,800 |

## G  REVERSE ENGINEERING SECRET KEY

Neural Cleanse (Wang et al., 2019) detects whether an image classifier is backdoored or not by reverse engineering backdoor triggers. Specifically, suppose we have an image classifier $h$ which is trained for a $C$-class classification task. Neural Cleanse views each class $c$, where $c = 1, 2, \cdots, C$, as a potential target class and reverse engineers a backdoor trigger for it. Suppose $\mathbf{\Delta}_c$ and $\mathbf{m}_c$ are the trigger pattern and trigger mask of the reverse engineered trigger, respectively. Formally, Neural Cleanse aims to find them by solving the following optimization problem:

$$\min_{\mathbf{m}_c, \mathbf{\Delta}_c} \mathcal{L} = \frac{1}{|\mathbf{X}|} \cdot \sum_{\mathbf{x} \in \mathbf{X}} CL(c, h((1 - \mathbf{m}_c) \cdot \mathbf{x} + \mathbf{m}_c \cdot \Delta_c)) + a \cdot |\mathbf{m}_c|, \quad (7)$$

where $CL$ is the cross-entropy loss, $\mathbf{X}$ is a set of clean images, $(1 - \mathbf{m}) \cdot \mathbf{x} + \mathbf{m} \cdot \Delta$ is an operation to inject the trigger into the example $\mathbf{x}$, and $a$ is the weight to penalize the mask size of the backdoor trigger. Neural Cleanse then obtains the $\ell_1$ norms of the reverse engineered triggers and applies an outlier detection method. If there exists at least one outlier, it treats $h$ as a backdoored classifier. Otherwise, it predicts $h$ as a clean one.

We generalize Neural Cleanse as an adaptive attack to reverse engineer the secret key of a locked foundation model. In particular, given a locked foundation model $f'$ and a downstream classification task, our goal is to reverse engineer the secret key (denoted by $\mathbf{k}_r$) such that the locked foundation model can achieve good performance on the downstream classification task with $\mathbf{k}_r$. In other words, our goal is to find $\mathbf{k}_r$ such that an input embedded with $\mathbf{k}_r$ is correctly predicted by a downstream classifier for the downstream task. As the goal of finding $\mathbf{k}_r$ is different from reverse engineering a backdoor trigger, we make the following modifications for Neural Cleanse to find $\mathbf{k}_r$. First, we remove the regularization term (second term in Equation 7) because the $\ell_1$-norm of the true secret key could be very large. The secret key is different from a backdoor trigger which is usually very small to stay stealthy. Second, we replace $(1 - \mathbf{m}_c) \cdot \mathbf{x} + \mathbf{m}_c \cdot \Delta_c$ with $\mathbf{x} \oplus \mathbf{k}_r$. The reason is that embedding the secret key into an input is different from embedding a backdoor trigger to the input. $\mathbf{x} \oplus \mathbf{k}_r$ aligns with how we embed the secret key into an input. Note that we also clip each entry of $\mathbf{x} \oplus \mathbf{k}_r$ to the range $[0, 1]$ following our key embedding operation. Formally, we formulate the following optimization problem to find $\mathbf{k}_r$:

$$\min_{\mathbf{k}_r} \mathcal{L} = \frac{1}{|\mathcal{D}_{dt}|} \cdot \sum_{(\mathbf{x}, \mathbf{y}) \in \mathcal{D}_{dt}} CL(\mathbf{y}, g(f'(\mathbf{x} \oplus \mathbf{k}_r))), \quad (8)$$

where $f'$ is the locked foundation model, $g$ is the downstream classifier, and $\mathcal{D}_{dt}$ is the downstream training dataset. We need to train a downstream classifier $g$ to solve the optimization problem. We iterative update $\mathbf{k}_r$ and train the downstream classifier $g$. Specifically, we first train a downstream classifier $g$ on $\mathcal{D}_{dt}$ based on the locked foundation model with the secret key $\mathbf{k}_r$. Then, we use SGD to update $\mathbf{k}_r$ based on Equation 8. Note that $\mathbf{k}_r$ is randomly initialized in the first round. We iterate until convergence.

**Table 7: Performance of FMLock on vision foundation models.**

| Model | Downstream dataset | ACC-baseline (%) | ACC-lock with key (%) | ACC-random (%) | ACC-lock without key (%) |
|---|---|---|---|---|---|
| CLIP | CIFAR10 | 86.06 | 85.46 | 17.82 | 20.17 |
| | STL10 | 96.59 | 95.14 | 16.10 | 15.76 |
| | SVHN | 56.80 | 56.91 | 19.59 | 19.59 |
| | EuroSAT | 83.52 | 82.30 | 15.41 | 12.74 |
| | GTSRB | 53.61 | 52.43 | 5.94 | 6.63 |

**Table 8: Impact of shadow dataset on FMLock. The shadow dataset size is 50,000. The locked foundation model is pre-trained on ImageNet.** *a subset of the pre-training dataset* **means the shadow dataset examples are sampled from ImageNet training set;** *same dist. as the pre-training dataset* **means the shadow dataset examples are sampled from ImageNet validation set;** *diff. dist. as the pre-training dataset* **means the shadow dataset examples are sampled from STL10 unlabeled data.**

| Downstream dataset | Shadow dataset | ACC-lock with key (%) | ACC-lock without key (%) |
|---|---|---|---|
| CIFAR10 | a subset of the pre-training dataset | 86.75 | 29.32 |
| | same dist. as the pre-training dataset | 87.56 | 29.04 |
| | diff. dist. as the pre-training dataset | 86.58 | 30.62 |
| STL10 | a subset of the pre-training dataset | 92.12 | 22.62 |
| | same dist. as the pre-training dataset | 92.09 | 23.00 |
| | diff. dist. as the pre-training dataset | 91.13 | 22.04 |
| SVHN | a subset of the pre-training dataset | 62.85 | 18.00 |
| | same dist. as the pre-training dataset | 63.12 | 19.58 |
| | diff. dist. as the pre-training dataset | 62.20 | 20.63 |
| EuroSAT | a subset of the pre-training dataset | 86.96 | 20.85 |
| | same dist. as the pre-training dataset | 87.07 | 19.19 |
| | diff. dist. as the pre-training dataset | 86.35 | 20.63 |
| GTSRB | a subset of the pre-training dataset | 50.80 | 5.95 |
| | same dist. as the pre-training dataset | 49.06 | 6.06 |
| | diff. dist. as the pre-training dataset | 49.01 | 6.82 |