# OpenReview forum: "FMLock: Preventing Unauthorized Use of Large Foundation Models"
_ICLR.cc/2024/Conference — ICLR 2024 Conference Withdrawn Submission_

### Official Review · Reviewer_QwY5 · 2023-10-25

**Soundness:** 2 fair
**Presentation:** 2 fair
**Contribution:** 2 fair
**Rating:** 6
**Confidence:** 3

**Summary:**

In this paper, the author proposed FMLock to prevent unauthorized use of large foundation models. Specifically, they used two algorithms KEYGEN() and MODELLOCKING(f,k) that satisfies utility-preserving and functionality-constraining properties on various types of foundation models.

**Strengths:**

1, the authors presents an interesting framework to lock a foundation model that only produces high-resolution output for authorized users with a secret key.

2, experimental restuls are promising.

3, considered more advanced adaptive attacks other than the random guesses of keys.

**Weaknesses:**

The presentation can be improved.

In particular, section 4 theoretical analysis needs more clarity. The analysis of obtaining theorem 3 is very confusing, and the proof for theorem 3 is basically a reiteration of the analysis in the paper. Maybe stating theorem 3 right after theorem 2 and explain the analysis after.

**Questions:**

How would this model perform under any kind of distribution shift?

---

### Official Review · Reviewer_zZJV · 2023-10-28

**Soundness:** 3 good
**Presentation:** 3 good
**Contribution:** 4 excellent
**Rating:** 8
**Confidence:** 3

**Summary:**

This paper proposes a secret-key-based approach to protect large foundation models. The approach is based on two loss functions which 1) improve the similarity between the original output and the output when the prompt is appended with the secret key. 2) decrease the similarity between the original output and the output when the prompt does not contain the key.

**Strengths:**

+ The problem studied in this work is timely and important.
+ The approach is straightforward and good.
+ The paper also incorporates a theoretical analysis part to prove the correctness and the security of this approach.

**Weaknesses:**

- The experimental part may provide more examples and cases to further provide the readers with more intuition on the effectiveness of the proposed approach on three scenarios.

**Questions:**

Please see the weakness part above.

---

### Official Review · Reviewer_r1iw · 2023-10-30

**Soundness:** 2 fair
**Presentation:** 3 good
**Contribution:** 2 fair
**Rating:** 3
**Confidence:** 4

**Summary:**

This paper introduces FMLock, a framework designed to secure foundation models across various domains including vision, language, and text-to-image models, ensuring optimal utility for authorized users while significantly constraining functionality for unauthorized users. The approach is based on a two-part mechanism comprising KEYGEN for secret key generation and MODELLOCKING for embedding the key into the model. Different procedures are tailored for each type of foundation model to cater to their unique characteristics. Experimental evaluations and theoretical analyses confirm FMLock's utility-preserving and functionality-constraining properties, demonstrating its resilience even in the face of adaptive attacks such as fine-tuning and reverse engineering attempts. The balanced approach in managing the two loss functions solidifies FMLock’s position as a solution for securing foundation models, providing a means of protection against unauthorized access while maintaining performance integrity for legitimate users.

**Strengths:**

1. The paper introduces a simple yet effective framework, FMLock, for securing foundation models through a unique combination of key generation (KEYGEN) and model locking (MODELLOCKING), protecting the model from unauthorized use.

2. The FMLock is designed to work across a broad range of foundation models, including vision, language, and text-to-image models, showcasing versatility and wide applicability.

3. FMLock adeptly maintains the foundation model's performance for authorized users with the secret key, while robustly securing it against unauthorized access, ensuring both functionality and protection.

**Weaknesses:**

1. As the number of users or applications expands, managing access through secret keys becomes increasingly cumbersome, potentially causing scalability issues and straining computational and memory resources, especially in environments where resources are limited.

    Specifically, the requirement for the model owner to continuously (re)train the foundational model to integrate a distinct secret key for every additional user continuously consumes huge computation resources and presents a considerable financial burden for the model owner.


2. The effectiveness of FMLock against real-world attackers remains to be thoroughly tested and validated in practical settings. For example, an adversary could pretend as a legitimate user to obtain a unique secret key. Possessing this key enables him to utilize the model, facilitating malicious activities such as constructing a high-fidelity surrogate model using high-quality input-output pairs, resulting in a potential financial losses for the original model owner.

    Although there could exist a CA to verify the identify of the user, the proposed design cannot detect the actions of model stealing (as this can be done at the local side of the adversary). Using a secret key to lock the model cannot solve the foundational problems of authorized use or model stealing.


3. The evaluation of FMLock is limited to accuracy and FID scores, neglecting other important metrics like efficiency, the time cost of tuning the model with a secret key and the cost of performing a prediction with/without the key.


4. The paper neglects to explore the implications of FMLock on regular model updating, which is very important as foundation models are constantly updated and improved upon.

**Questions:**

1. How does FMLock handle the complexity and potential vulnerabilities in key management, especially in large-scale environments?

2. How does FMLock plan to mitigate the additional computational and memory resources required for locking models, particularly in resource-constrained settings?

3. How des FMLock prevent real-world adaptive attacks (refer to the example (i.e., the malicious user) depicted in the Weaknesses Section)?

4. How does FMLock accommodate routine model updating and maintenance, ensuring consistent security enforcement across different model versions?

---

### Official Review · Reviewer_4dyf · 2023-11-01

**Soundness:** 2 fair
**Presentation:** 2 fair
**Contribution:** 1 poor
**Rating:** 3
**Confidence:** 4

**Summary:**

This work considers the problem of locking foundation models to prevent malicious use. The major contributions of this work are:

1) FMLock, a framework to lock foundation models that can only be used by actors with access to a secret key.

2) Experiments on both discriminative foundational models (CLIP) and generative foundational models (BERT, Stable-Diffusion) demonstrate acceptable results.

**Strengths:**

1) This paper is easy to follow.

**Weaknesses:**

1) Practicality of the Contribution: The feasibility of preventing malicious use of foundation models via the proposed solution raises many concerns. For instance:

- An authorized user could distill new foundation or specialist models from the locked model and distribute them, inadvertently enabling access for malicious actors.
- There exists a risk that an authorized user might exploit the model to deduce additional secret keys and disseminate them, potentially leading to unauthorized and harmful use.

2) Ambiguity in defining Malicious/ unauthorized use: The paper lacks clarity in defining what constitutes "malicious use," leading to ambiguity in its thesis and overall scope. A robust defense mechanism against adversarial misuse might be a more viable approach than restricting model access entirely.

3) Breadth of Focus: The inclusion of both discriminative and generative foundation models, while comprehensive, dilutes the focus. For generative models, existing literature on watermarking LLMs [A] and diffusion models for image generation [B, C] could offer valuable insights. A discussion on these methodologies would improve the paper’s relevance.

4) Empirical results. The empirical results are not surprising, given the optimization objective in Equation (3). More details on the shadow dataset for different setups should be included. My understanding is that this work finetunes CLIP, BERT, Stable-Diffusion using Equation (3) and the shadow dataset. Can the authors confirm this?

5) Scalability issues. FMLock scales with the number of keys. This is a serious drawback in the proposed scheme, i.e., If one anticipates 1M “authorized” users, one needs to train the system with 1M unique keys.

6) Error bars are missing for all the experiments.

Overall I enjoyed reading this paper and appreciate this new effort. But in my opinion, the weaknesses of this paper significantly outweigh the strengths. But I’m willing to change my opinion based on the rebuttal.


[A] Kirchenbauer, J., Geiping, J., Wen, Y., Katz, J., Miers, I. &amp; Goldstein, T.. (2023). A Watermark for Large Language Models. ICML 2023.

[B] Wen, Yuxin, et al. "Tree-Ring Watermarks: Fingerprints for Diffusion Images that are Invisible and Robust." arXiv preprint arXiv:2305.20030 (2023).

[C] Zhao, Yunqing, et al. "A recipe for watermarking diffusion models." arXiv preprint arXiv:2303.10137 (2023).

**Questions:**

Please see Weaknesses section above for a list of all questions.

---

### Official Review · Reviewer_hoDc · 2023-11-04

**Soundness:** 2 fair
**Presentation:** 3 good
**Contribution:** 2 fair
**Rating:** 3
**Confidence:** 5

**Summary:**

This paper studies the problem of preventing unauthorized usage of foundation models by injecting some secret keys into the model training. Only for inputs with the secret keys (designed by the users), the foundation model will produce desired high-quality outputs and for inputs without secret keys, the quality of the outputs will extremely low. This approach is known as "locking" the foundation model. Empirical results on multiple language-vision foundation models and vision language models show the proposed approach can achieve the desired locking functionality.

**Strengths:**

1. The attempt is preventing unauthorized usage of foundation models is important for the community.
2. The presentation is mostly clear and is easy to follow.

**Weaknesses:**

1. The application scenario of the proposed approach is unclear and in fact, might be fairly limited. It assumes the model owner distributes its own secret key (implicit assumption of using only 1 secret key) but does not discuss how to distinguish between authorized and unauthorized parties. It is highly likely that adversaries can pretend to be authorized parties to obtain the secret key, or some authorized parties can leak the secret key (similar argument is made in the paper to show why hosting foundation models as a cloud service is not a solution). Because the secret key is shared by all users, once the secret key is (easily) obtained, the purpose of the model is completely defeated. The "secret key" discussed in this paper is different from the "secret key" we usually discuss in data encryption.
2. The downstream fine-tuning shows upward trajectory when the number of downstream samples increases, which makes me wonder if all the downstream samples are used in fine-tuning or reverse engineering, if the proposed method will be completely broken. The authors argued that there cannot be many downstream samples, otherwise the adversary may choose to train a model from scratch. However, there is still strong incentives for adversaries to use pretrained models even with some sufficient downstream samples because 1), the available data points may still not be sufficient to train very large models (while feasible through transfer learning), and 2) the adversaries may not have the computational resources to perform from-scratch training.

**Questions:**

1. The implicit assumption in the paper is the model owner only uses one secret key. Will the proposed approach fail when there are many (e.g., thousands or more) secret keys?
2. I do not understand the motivation behind performing ablation studies on the shadow dataset? The model owner has access to the shadow dataset and therefore, can always opt to use the more representative shadow dataset from the same distribution.